

# Canopy soil bacterial communities altered by severing host tree limbs

Cody R. Dangerfield, Nalini M. Nadkarni and William J. Brazelton

Department of Biology, University of Utah, Salt Lake City, UT, United States of America

## ABSTRACT

Trees of temperate rainforests host a large biomass of epiphytic plants, which are associated with soils formed in the forest canopy. Falling of epiphytic material results in the transfer of carbon and nutrients from the canopy to the forest floor. This study provides the first characterization of bacterial communities in canopy soils enabled by high-depth environmental sequencing of 16S rRNA genes. Canopy soil included many of the same major taxonomic groups of Bacteria that are also found in ground soil, but canopy bacterial communities were lower in diversity and contained different operational taxonomic units. A field experiment was conducted with epiphytic material from six *Acer macrophyllum* trees in Olympic National Park, Washington, USA to document changes in the bacterial communities of soils associated with epiphytic material that falls to the forest floor. Bacterial diversity and composition of canopy soil was highly similar, but not identical, to adjacent ground soil two years after transfer to the forest floor, indicating that canopy bacteria are almost, but not completely, replaced by ground soil bacteria. Furthermore, soil associated with epiphytic material on branches that were severed from the host tree and suspended in the canopy contained altered bacterial communities that were distinct from those in canopy material moved to the forest floor. Therefore, the unique nature of canopy soil bacteria is determined in part by the host tree and not only by the physical environmental conditions associated with the canopy. Connection to the living tree appears to be a key feature of the canopy habitat. These results represent an initial survey of bacterial diversity of the canopy and provide a foundation upon which future studies can more fully investigate the ecological and evolutionary dynamics of these communities.

# INTRODUCTION

Temperate wet forests support a large biomass and high diversity of epiphytic plants (*Van Pelt et al., 2006*; *Tejo, Zabowski & Nadkarni, 2015*; *Haristoy, Zabowski & Nadkarni, 2014*; *Nadkarni, 1984b*; *Nadkarni, 1984a*; *Ellyson & Sillett, 2003*) that are accompanied by extensive accumulations of organic canopy soils up to 30 cm thick (*Haristoy, Zabowski & Nadkarni, 2014*). Single trees can support over 6.5 t dry weight of live and dead epiphytic material (EM), nearly four times the foliar biomass of host trees (*Nadkarni, 1984a*). These canopy communities play important ecological roles in ecosystem processes, particularly in whole-forest nutrient cycles. Epiphytic plants are supported by their host trees but acquire nutrients mainly from atmospheric sources (precipitation and particulates that settle within

Corresponding author
William J. Brazelton,
william.brazelton@utah.edu

or move through the canopy) (*Prescott, 2002*; *Pérez et al., 2005*; *Nadkarni & Solano, 2002*). Canopy soils develop from the accumulation and decomposition of EM on branches and in bifurcations of trees (*Pérez et al., 2005*; *Nadkarni & Solano, 2002*; *Enloe & Graham, 2006*). Canopy soils retain water and nutrients in their airspaces and on surface exchange sites, respectively (*Nadkarni, 1981*; *Lindo & Whiteley, 2011*). When EM falls from branches or "rides down" with broken branches or fallen trees, these nutrients can be transferred to the forest floor and become available to terrestrial vegetation as they die and decompose. Additionally, some host trees gain access to the nutrients in EM directly via canopy roots (*Nadkarni, 1981*). EM also creates habitat for birds, invertebrates, and arboreal mammals (*Nadkarni, 1981*; *Coxson & Nadkarni, 1995*; *Pypker & Unsworth, 2006*; *Wolf, 2005*).

Most studies of EM have focused on their diversity, the pools of nutrients they store, or the ecosystem services they provide. However, little information exists on the biota and processes responsible for the dynamics of EM as it moves from canopy to the forest floor. Epiphytes attached to a fallen tree or branch on the ground may remain vigorous for some time, but the chances for survival of those fallen to the shady ground are low (*Matelson, Nadkarni & Longino, 1993*). The rates, processes, and biota responsible for their death and decomposition have been documented in only a few tropical forests (*Cabral et al., 2015*; *Zotz et al., 1998*; *Schmidt & Zotz, 2000*; *Hietz, Buchberger & Winkler, 2006*; *Aubrey, Nadkarni & Broderick, 2013*; *Kozich et al., 2013*; *Edgar, 2010*; *Schloss et al., 2009*). This information is critical to understand the biology and ecological roles of the living communities and their accompanying soils in whole-ecosystem processes.

To explore the microbial ecology of canopy soil, we carried out an experiment in a temperate rainforest in Olympic National Park, Washington, USA. The field experiment was designed to investigate the effects of disturbance and movement of EM from the canopy to the forest floor on the resident bacterial communities, which are presumably associated with the decline and decomposition of EM. We compared bacterial community diversity and composition in EM samples that (1) were located on living vs. dead branch substrates and (2) experienced canopy vs. forest floor environments. These results were used to identify those bacterial taxa that were resident to the canopy soil and those that colonized the transplanted canopy material from other sources.

## MATERIALS AND METHODS

### Site description

The study was conducted in the Upper Quinault River Valley of the Olympic National Park, Washington, USA (47.52°N 123.82°W). Research permits were issued from the Research Office at the Olympic National Park (OLYM-000234). Average annual precipitation is ∼350 cm in the lowlands and ∼510 cm in the higher elevations. The fall, winter, and spring are characterized by heavy rains; summers are typically dry (*Aubrey, Nadkarni & Broderick, 2013*). The floodplain forest of the study area is predominated by Big-leaf Maple (*Acer macrophyllum*), which supports the largest epiphyte loads. Other tree species present are Sitka spruce (*Picea sitchensis*), red alder (*Alnus rubra*), and Douglas-fir (*Pseudotsuga menziesii*). Epiphytic material (EM) in Big-leaf maple is described by

*Aubrey, Nadkarni & Broderick (2013)* and consists of live epiphytes that overlie a thick layer of arboreal soils. Live epiphytes (mosses, liverworts, lichens, and licorice fern, *Polypodium glycyrrhiza)* are dominated by two bryophyte species, *(Isothecium myosuroides* and *Antitrichia curtipendula)*. Accumulations of arboreal soils are greatest in branch bifurcations at the trunk (up to 30 cm thick), and taper to small amounts at branch tips. These soils are composed of dead and decomposing epiphytes that remain on host tree branches, and small amounts of intercepted host tree litter. Arboreal and forest floor soil characteristics are described in Tejo et al. (*Haristoy, Zabowski & Nadkarni, 2014*).

## Sample collection

On September 28, 2012 we selected nine *A. macrophyllum* trees within three previously established research plots (3 ha each, within 7 km$^2$ from each other) (*Aubrey, Nadkarni & Broderick, 2013*). See File S1 for photographs of trees and samples. Criteria for inclusion were: safe canopy accessibility; no apparent dead or diseased branches; no visual access from the National Park road; multiple potential sampling branches; mature status; large loads of live epiphytes; and a minimum distance of 200 m from each other. Three of the trees were designated as ''experimental trees'', onto which the experimental treatments were transplanted during the experiment. Six others were designated as ''source trees'' from which samples were collected for the experiment. From these source trees, 13 branches (6–10 cm in diameter, 11–18 m from the ground) with complete epiphyte cover were selected, cut, and lowered to the forest floor by an arborist. These branches were cut into 75 cm length segments and labeled. These severed segments were then randomly selected to one of following treatments within and beneath the experimental trees (Figs. 1 and 2): (A) suspended within the canopy (canopy-severed) at the same height, (B) placed below on the forest floor (ground-perched), or (C) EM was stripped and placed directly on the forest floor (ground-flat). Canopy soil and soil from the stripped branches from each segment were sampled by first removing the overlying live epiphytic material from the surface of the epiphyte mats, and then retrieving samples (ca. $2 \times 2 \times 2$ cm) from soil 5 cm below the canopy soil surface.

Two years later (September 14, 2014), we sampled canopy soil from all treatments as well as from undisturbed EM in the canopy of experimental trees (canopy-original) and from forest floor soil (ground-original) from locations that were randomly located beneath the crown, between the trunk and drip line of each of the experimental trees (Fig. 1). For forest floor samples, the overlying leaf litter was removed, and samples (ca. $2 \times 2 \times 2$ cm) were taken from 5 cm below the forest floor surface. Both canopy and forest floor soils appeared to be homogenous at that depth. The effect of host tree was evaluated by repeating the bacterial diversity analyses described below after categorizing samples by host tree rather than by experimental treatment. No trends specific to any of the host trees were observed.

## Extraction of DNA from soil samples

The samples were homogenized for DNA extraction by flash-freezing the sample with liquid nitrogen followed by grinding the sample into a fine powder with a sterilized mortar and pestle. DNA was extracted from each sample using the PowerSoil DNA Isolation Kit

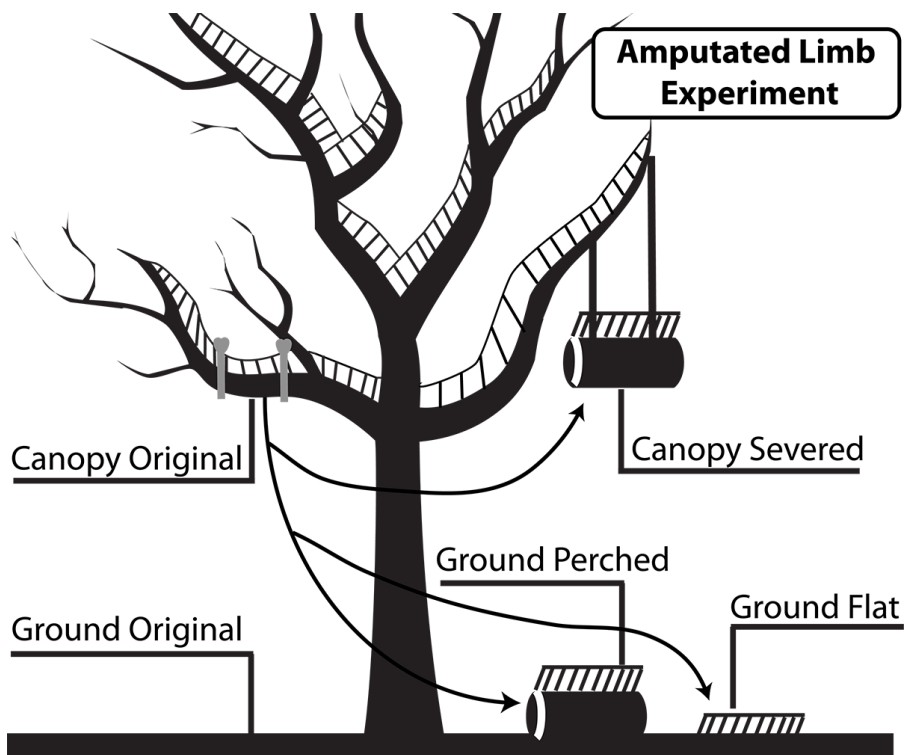

**Figure 1** **Experimental design to investigate effects of disturbances on canopy soil bacterial communities.** The undisturbed canopy soil attached to live branches (canopy-original) was compared to three experimental treatments: canopy epiphytic material (EM) on severed dead branches suspended in the canopy (canopy-severed), canopy EM on dead branches transplanted to the forest floor (ground-perched), and canopy EM removed from the branch and placed directly on the forest floor (ground-flat). In addition, all treatments were compared to undisturbed ground soil underneath the tree (ground-original). Illustration by Renae Curtz and Doug Cornwall.

(MO BIO Laboratories, Carlsbad, CA, USA) according to the manufacturer's instructions and stored at −20 °C.

## Bacterial 16S rRNA gene sequencing

Bacterial 16S rRNA gene amplicon sequencing was conducted by the Michigan State University genomics core facility. The V4 region of the 16S rRNA gene (defined by primers 515F/806R) was amplified with dual-indexed Illumina fusion primers as described by *Kozich et al. (2013)*. Amplicon concentrations were normalized and pooled using an Invitrogen SequalPrep DNA Normalization Plate. After library QC and quantitation, the pool was loaded on an Illumina MiSeq v2 flow cell and sequenced using a standard 500 cycle reagent kit. Base calling was performed by Illumina Real Time Analysis (RTA) software v1.18.54. Output of RTA was demultiplexed and converted to fastq files using Illumina Bcl2fastq v1.8.4. Paired-end sequences were filtered and merged with USEARCH 8 (*Edgar, 2010*), and additional quality filtering was conducted with the mothur software platform (*Schloss et al., 2009*) to remove any sequences with ambiguous bases and more than 8 homopolymers. Chimeras were removed with mothur's implementation of UCHIME (*Edgar et al., 2011*). The sequences were pre-clustered with the mothur command pre.cluster (diffs =1), which

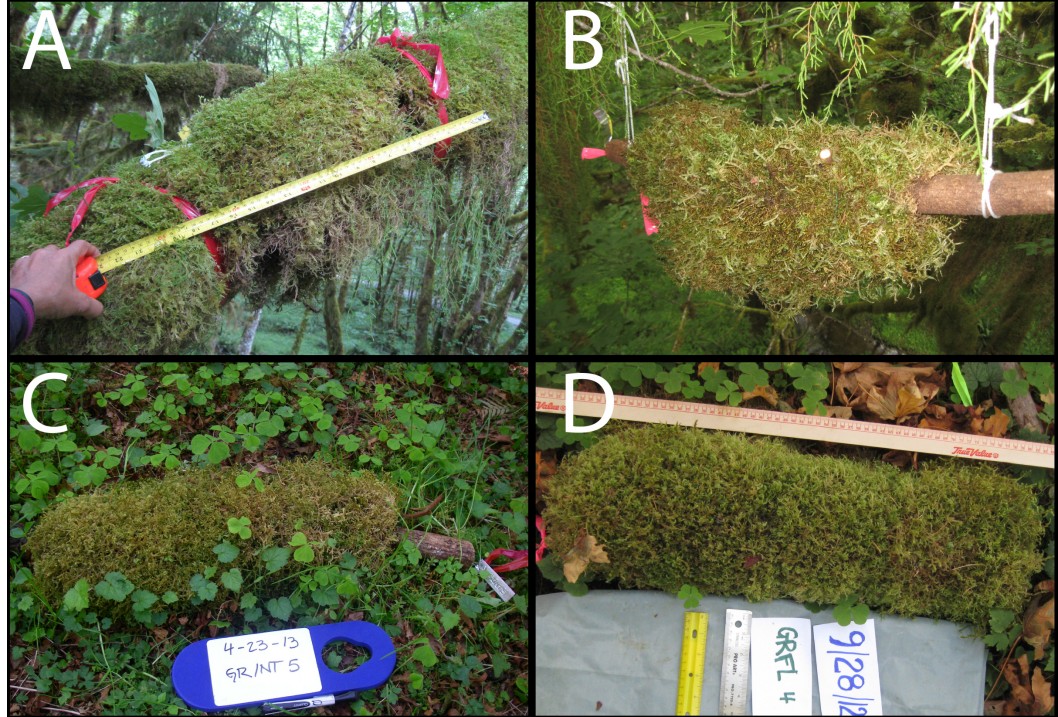

**Figure 2** **Photographs of canopy samples described in Fig. 1.** (A) Canopy-original samples consisted of undisturbed areas of epiphyte mats (EM), marked at both ends. (B) Canopy-severed samples were branch segments taken from nearby source trees; segments with intact EM were hung with strings below branches that were adjacent to the canopy-original samples. (C) Ground-perched samples were placed on the ground, with intact EM on branch substrates. (D) Ground-flat samples were placed on the ground, with EM removed from their branch substrate and laid flat to have direct contact on the forest floor. Ground-original treatments (not pictured here) were collected from the forest floor, with the O-layer removed. All photos were taken at the beginning of the experiment (28 Sept 2012) except for ground-perched (C), which was photographed on 23 April 2013. Photo credit: N Nadkarni.

reduced the number of unique sequences from 574,178 to 351,566. This pre-clustering step removes rare sequences most likely created by sequencing errors (*Schloss & Westcott, 2011*).

## Bacterial diversity analyses

These unique, pre-clustered sequences were considered to be the operational taxonomic units (OTUs) of this study and formed the basis of all alpha and beta diversity analyses. We chose not to cluster sequences any more broadly because clustering inevitably results in a loss of biological information and because no arbitrary sequence similarity threshold can be demonstrated to consistently correspond to species-like units. Samples with fewer than 20,000 sequences (15 of the original 52 samples) were considered to have failed the sequencing step and were removed from analysis. The removed samples were roughly equally distributed among the controls and treatments, and choosing lower or higher thresholds (i.e., removing more or fewer samples) did not substantially alter any results. All 37 high-quality samples were randomly sub-sampled down to 20,259 sequences prior to calculation of richness, evenness, and alpha diversity. Taxonomic classification of all sequences was performed with mothur using the SILVA reference alignment (SSURefv123)

and taxonomy outline (*Pruesse, Peplies & Glöckner, 2012*). Taxonomic counts generated by mothur and edgeR results were visualized in bar charts generated with the aid of the R package phyloseq (*McMurdie & Holmes, 2013*). Diversity analyses were repeated after removing all sequences identified as mitochondrial or chloroplast 16S rRNA, but this procedure did not substantially affect any results, as these sequences are a minor component of the overall species richness. Therefore, mitochondria and chloroplast sequences were retained in the presented analyses because of their potential added value in aiding ecological interpretations.

## Statistical analyses

Alpha diversity and evenness were calculated with the invsimpson and simpsoneven calculators in the mothur package, and the standard error of the mean of samples within each experimental treatment are reported in Table 1. Differences between observed numbers of OTUs and evenness in each treatment were tested for significance with a Dunnett-Tukey-Kramer test, which accounts for multiple comparisons among samples with unequal sizes and variances (*Lau, 2013*). The dissimilarity of bacterial community compositions (i.e., beta diversity) was calculated with the Morisita-Horn index from a table of OTU abundances across all samples. This index was chosen because it reflects differences in the abundances of shared OTUs without being skewed by unequal numbers of sequences among samples. Morisita-Horn community dissimilarity among samples was visualized with a multi-dimensional scaling (MDS) plot generated with the distance, ordinate, and plot ordination commands in phyloseq. The ggplot2 function stat_ellipse was used to draw 95% confidence level ellipses (assuming t-distribution) on Fig. 3 for each group containing more than five samples. Additionally, the significance of community composition differences between groups of samples was calculated with an AMOVA (analysis of molecular variance) as implemented in mothur (*Pruesse, Peplies & Glöckner, 2012*) and reported in Table 2. Differences in the relative abundances of OTUs between experimental treatments were measured with the aid of the R package edgeR (*Robinson, McCarthy & Smyth, 2010*) as recommended by McMurdie and Holmes (*McMurdie & Holmes, 2014*). The differential abundance of an OTU (as measured in units of $\log_2$ fold change) was considered to be statistically significant if it passed a false discovery rate threshold of 0.05. OTUs were assigned to canopy or ground soil sources using the sink-source Bayesian approach of SourceTracker2 v2.0.1 (https://github.com/biota/sourcetracker2) with rarefying to 20,000 sequences for sinks and sources (*Knights et al., 2011*). Similar results were achieved without rarefying the data.

## Accession numbers

All sequence data are publicly available at the NCBI Sequence Read Archive under BioProject PRJNA357844. All SRA metadata, protocols, and supplementary datasets (including an interactive visualization of File S2 with Krona graphs) are archived at the following DOI: 10.5281/zenodo.597545. All custom software and scripts are available at https://github.com/Brazelton-Lab.

**Table 1** Average species richness and evenness between treatments.

| Sample type | $S_{OBS}$ | Inverse Simpson | Evenness (from Simpson) |
|---|---|---|---|
| Canopy-original | $9,705 \pm 838^a$ | $195 \pm 30$ | $0.022 \pm 0.004^{ab}$ |
| Canopy-severed | $9,464 \pm 651^{ab}$ | $194 \pm 40$ | $0.022 \pm 0.005^{ab}$ |
| Ground-perched | $13,680 \pm 790^b$ | $678 \pm 93$ | $0.050 \pm 0.008^{ab}$ |
| Ground- flat | $13,744 \pm 1,321^{ab}$ | $609 \pm 323$ | $0.042 \pm 0.019^{ab}$ |
| Ground-original | $13,168 \pm 493^b$ | $561 \pm 60$ | $0.043 \pm 0.006^{ab}$ |

**Notes.**

Observed species ($S_{OBS}$) in canopy-original samples (a) were significantly fewer than those in ground-perched and ground-original samples (b) according to a Dunnet–Tukey–Kramer test. All other comparisons (ab) were not significant. All values are reported $\pm$ standard error.

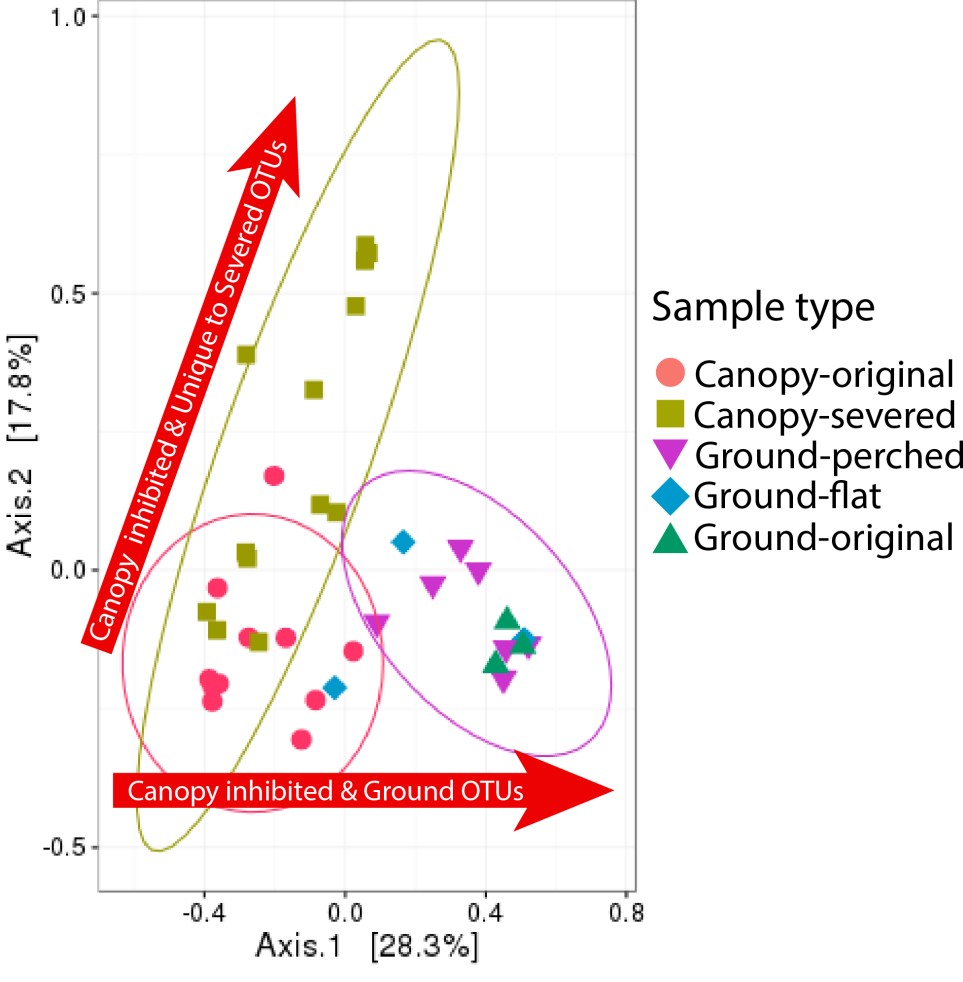

**Figure 3** Shifts in bacterial community composition associated with canopy-severed compared to ground-perched and ground-flat treatments with canopy-original and ground-original representing the original community compositions. The ellipses indicate where 95% of samples within a treatment are expected to occur on the plot. Ellipses could only be drawn for sample types containing at least five samples. Arrows reflect the interpretations of which taxa are affected by each treatment, as described in the text.

**Table 2** AMOVA analysis of significant differences in bacterial community compositions.

| Description | Comparison | $F_s$ | *p*-value |
|---|---|---|---|
| Overall | All 5 sample groups | 5.5807 | <0.001 |
| Undisturbed canopy and ground soil | CA-OR vs. GR-OR | 7.42532 | 0.003 |
| Treatments compared to undisturbed canopy soil | CA-SE vs. CA-OR | 5.7236 | <0.001 |
| | GR-PE vs. CA-OR | 10.2246 | <0.001 |
| | GR-FL vs. CA-OR | 3.18728 | 0.003 |
| Treatments compared to undisturbed ground soil | CA-SE vs. GR-OR | 6.75715 | <0.001 |
| | GR-PE vs. GR-OR | 3.74506 | 0.013 |
| | GR-FL vs. GR-OR | 1.35554 | 0.188 |
| Comparisons between treatments | CA-SE vs GR-FL | 3.16961 | 0.004 |
| | CA-SE vs. GR-PE | 8.20981 | <0.001 |
| | GR-FL vs. GR-PE | 1.90745 | 0.063 |

**Notes.**

Abbreviations: CA-OR, canopy-original; CA-SE, canopy-severed; GR-PE, ground-perched; GR-FL, ground-flat; GR-OR, ground-original.

# RESULTS

## Richness and evenness of soil bacterial communities

Table 1 lists the operational taxonomic unit (OTU) richness and evenness of bacterial communities inhabiting soil samples collected during this study. EM in the canopy (canopy-attached) had lower OTU richness compared to forest floor soil. Evenness values were lower, on average, in canopy samples compared to ground soil samples, but these differences were not statistically significant. Bacterial communities of EM on branches that were severed from the tree and suspended in the canopy (canopy-severed) had richness and evenness values that were indistinguishable from those of canopy-original samples. The OTU richness of bacterial communities of EM perched on branches that were moved to the forest floor (ground-perched) was significantly greater than that of the original canopy samples and was indistinguishable from the richness of ground soil samples.

## OTU composition of soil bacterial communities

At a broad taxonomic level, all samples from canopy and forest floor soils were generally similar, featuring roughly even representation of many bacterial groups commonly found in previously studied soils, including Rhizobiales, Acidobacteria, Actinobacteria, Sphingobacteria, Myxococcales, Xanthomonadales, and Verrucomicrobia. One notable exception is the order Nitrosomonadales (Betaproteobacteria), which was consistently 10–100 times less abundant in canopy compared to ground soils (File S2). At the level of individual OTUs, differences in bacterial community composition were more easily identified. For example, even though the order Rhizobiales (Alphaproteobacteria) was abundant in both canopy-original and ground-original samples, the most abundant Rhizobiales OTUs in canopy-original were not abundant in ground-original (and vice versa). This trend of similar abundances at the phylum, class, and order level but stark contrasts at the OTU level is evident for nearly all of the major taxonomic divisions of Bacteria in the soil samples (File S3).

In addition to having lower richness, canopy soils had significantly different OTU compositions compared to ground soils (Fig. 3 and Table 2). These differences are visualized in the MDS plot in Fig. 3, where each data point represents the OTU structure of one sample, and the distance between points represents the dissimilarity between samples. The three ground-original samples are clustered together in the MDS plot of Fig. 3 apart from the many canopy-original samples, and the AMOVA test reported in Table 2 confirms a significant difference between these sample groups.

Furthermore, the 95% confidence ellipse surrounding canopy-original samples in Fig. 3 almost entirely excludes all samples of canopy EM that had been transplanted to the ground on a severed branch (ground-perched). The OTU compositions of these ground-perched samples could not be distinguished from ground-original samples, as evidenced by the 95% confidence ellipse surrounding ground-perched samples also including all ground-original samples in Fig. 3 and by the non-significant AMOVA results in Table 2 (using a Bonferroni-corrected $p$-value threshold of 0.005).

Figure 3 also shows the variability in the bacterial community compositions of canopy-severed samples and an apparent gradient from the original canopy community composition to the most divergent canopy-severed community compositions. As a group, canopy-severed samples were distinct from canopy-original samples and all other treatments (Table 2). Furthermore, the shift in canopy-severed communities associated with severing the branch from the tree is distinct from the shift in ground-perched and ground-flat communities associated with transplanting the EM from the canopy to the ground (two arrows in Fig. 3).

## OTUs with differential abundance in canopy vs. ground soil

To identify individual OTUs that are significantly more abundant in canopy soil compared to ground soil (and vice versa), we contrasted the relative abundances of OTUs in canopy-original to the OTU abundances in ground-original samples. In Fig. 4, each data point represents the total abundance of each OTU across all samples ($X$-axis, in units of $\log_2$ counts per million sequences) and the differential abundance of each OTU between canopy-original and ground-original ($Y$-axis, in units of $\log_2$ fold change). Red data points represent OTUs whose differential abundances passed a significance test (false discovery rate <0.05) and can be thought of as the OTUs that are characteristic to that sample type. This analysis identified, for example, several Pseudomonadaceae OTUs that were more abundant in ground soil and nearly absent in the canopy (Fig. 4 and File S3). Furthermore, some OTUs classified as family Bradyrhizobiaceae (order Rhizobiales) were significantly more abundant in ground-original than in canopy-original. The Bradyrhizobiaceae also included other OTUs with the opposite abundance distribution; i.e., they were more abundant in canopy-original than in ground-original. In other words, canopy soil and ground soil each have their own distinct and abundant Bradyrhizobiaceae OTUs. A similar pattern was observed for the Acidobacteriaceae; some OTUs were significantly more abundant in ground soil, and other OTUs were more abundant in the canopy (Fig. 4 and File S3).

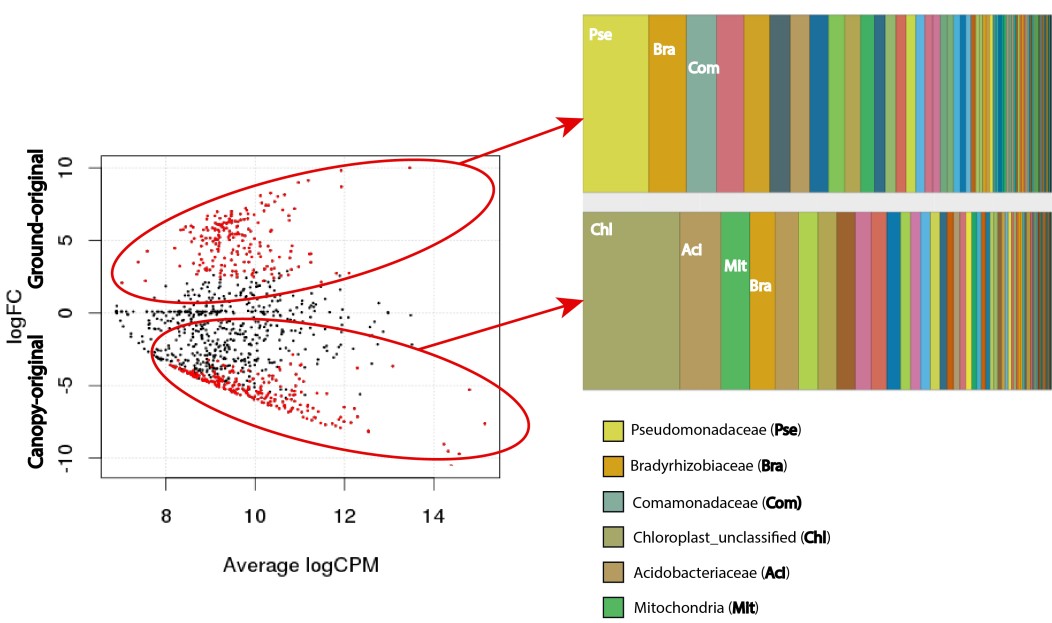

**Figure 4** **Differential abundance of OTUs in undisturbed canopy soil (canopy-original) and undisturbed ground soil (ground-original).** Red data points indicate OTUs with significantly greater abundance in canopy-original (lower half of plot) or ground-original (upper half of plot), as measured by $\log_2$ fold change (log FC). The *X*-axis shows the average abundance (sequence counts) of each OTU among all samples in the dataset in units of $\log_2$ counts per million (log CPM). Significance was defined as false discovery rate <0.05. Taxonomic classifications of OTUs with differential abundance in each sample type are provided as bar charts. Taxonomic groups with the most numbers of OTUs are labeled with abbreviations defined as bold text in the legend below.

Chloroplasts and mitochondria (both of which are detected by sequencing of bacterial 16S rRNA genes) were among the most common sources of differentially abundant OTUs between canopy-original and ground-original (Fig. 4). Most of the chloroplast sequences could not be classified because chloroplast 16S rRNA genes are not reliable taxonomic markers, but the best BLAST hits in the GenBank non-redundant database to the most abundant chloroplast sequences in canopy-original include those belonging to mosses and angiosperms as well as the lycopod *Selaginella.* The most abundant mitochondrial 16S rRNA sequences from canopy-original matched those of diverse ferns, the moss *Funaria hygrometrica,* and the lichenized fungus genus *Psora* (Fig. 4 and File S3). These taxonomic classifications are consistent with the organisms known to inhabit canopy EM.

## OTUs with differential abundance in experimental treatments

The abundance distribution pattern of each OTU was examined in order to identify the specific bacterial taxa driving the community shifts associated with experimental disturbances to canopy soil. Nearly all of the highly abundant OTUs were detected in most experimental treatments, but many of these OTUs had significantly greater abundances in one or more treatments compared to canopy-original (red data points in Fig. 5). There were 164 OTUs more abundant in canopy-severed compared to canopy-original (Fig. 5A), 245 OTUs that were more abundant in ground-perched compared to canopy-original

(Fig. 5B), and 196 OTUs more abundant in ground-flat compared to canopy-original (Fig. 5C). These differentially abundant OTUs must be primarily responsible for the shifts in community composition evident in Fig. 3.

Most OTUs that were highlighted by the differential abundance tests were found in multiple sample types. For example, 58% of the OTUs that were more abundant in canopy-severed compared to canopy-original had similar abundances in ground-perched, ground-flat, and ground-original (pie chart in Fig. 5A). Therefore, these OTUs are abundant everywhere except canopy-original and were designated 'Canopy Inhibited'. The remaining 42% of OTUs that were differentially abundant in canopy-severed compared to canopy-original were significantly less abundant or absent in all of the ground samples and were designated 'Unique to Canopy-Severed'.

Nearly all of the OTUs that were differentially abundant in ground-perched and ground-flat compared to canopy-original were found at similar abundances in nearby ground soil (ground-original). A few of these OTUs were the same OTUs identified as "Canopy Inhibited" above, and the remaining OTUs were designated as "Ground OTUs" (pie charts in Figs. 5B–5C), which are inferred to be derived from the nearby ground soil. Very few OTUs were uniquely abundant in the ground-perched or ground-flat treatments, which is consistent with the positions of ground-perched and ground-flat samples overlapping with those of canopy-original and ground-original samples in the MDS plot of Fig. 3.

## Taxonomic classifications of differentially abundant OTUs

In general, the differentially abundant OTUs included representatives from all of the typical soil taxonomic groups listed above and were not obviously divergent from the general community at broad taxonomic levels. A notable exception is that OTUs classified as family Acidobacteriaceae (phylum Acidobacteria) and family Acidothermaceae (phylum Actinobacteria) were much more abundant in canopy-original compared to any of the treatments (File S3).

The "Canopy Inhibited" and "Unique to Severed" categories of OTUs were also similar at broad taxonomic levels but differed at more specific taxonomic resolution (File S3). For example, all Rhizobiales OTUs that were more abundant in canopy-severed than canopy-original and classified as family Bradyrhizobiaceae (including genus *Bradyrhizobium*, which is typically found in plant root nodules) were identified as "Canopy Inhibited" because these sequences were also abundant in ground-original. In contrast, several unclassified Rhizobiales OTUs in canopy-severed were absent in ground soil and were therefore included in the "Unique to Severed" category. Within phylum Actinobacteria, OTUs in class Actinobacteria were overwhelmingly "Canopy Inhibited" while class Thermoleophilia were mostly "Unique to Severed". OTUs classified as Xanthomonadales were also found in both "Canopy Inhibited" and "Unique to Severed" categories. Chloroplast and mitochondria sequences with high abundance in canopy-severed were mostly absent in ground-original (and are therefore included in the "Unique to Severed" category), and many of these sequences were similar to those from mosses and liverworts (File S3). The "Canopy Inhibited" category also included many Chloroflexi OTUs (classes Anaerolineae and Ktedonobacteria).

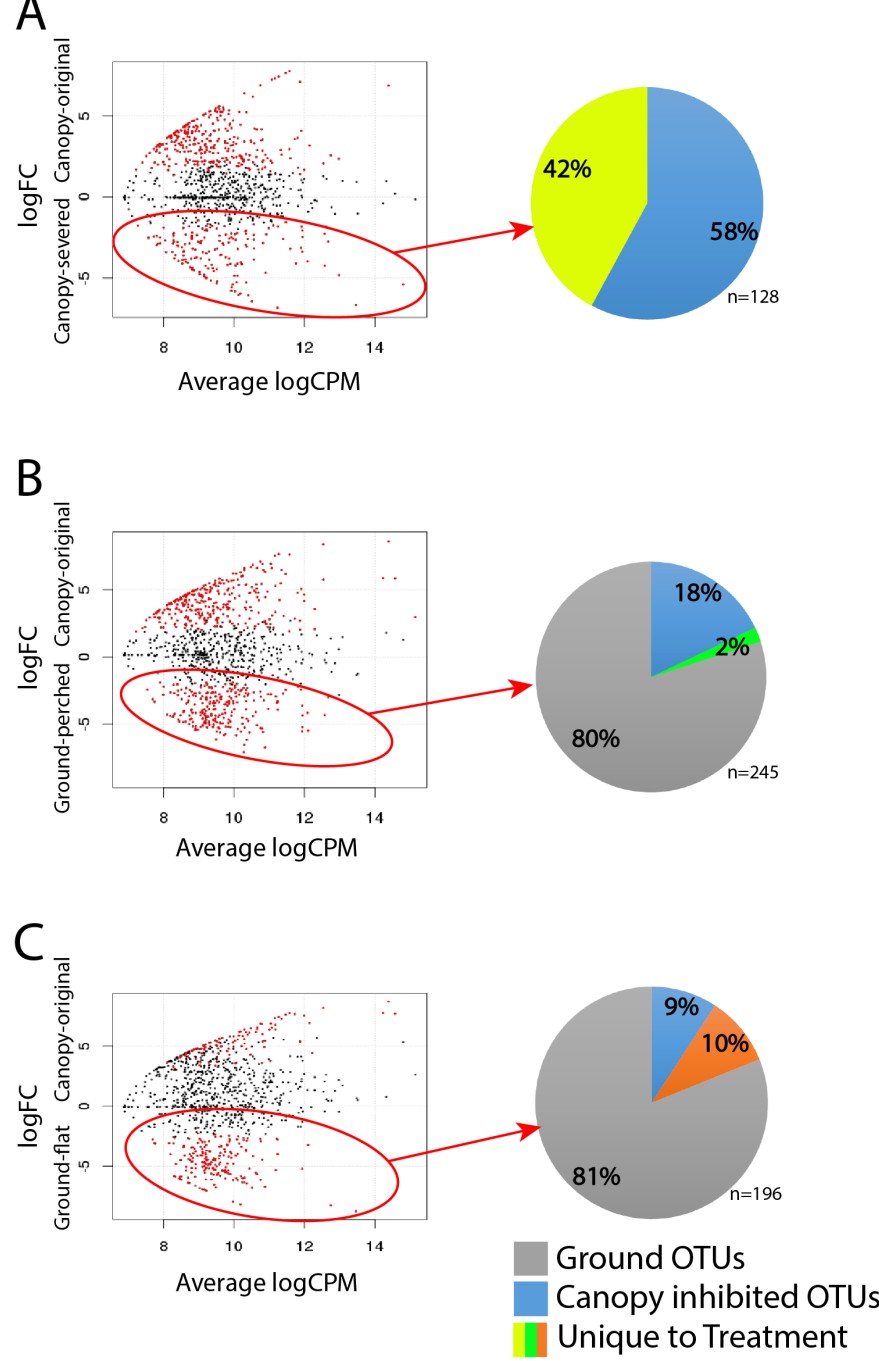

**Figure 5** **Differential abundance analysis to identify specific taxa with significantly greater abundance in one treatment compared to their abundance in undisturbed canopy soil: (A) canopy-severed vs. canopy-original, (B) ground-perched vs. canopy-original, (C) ground-flat vs. canopy-original.** Red data points indicate OTUs whose differential abundance passed a significance test (false discovery rate <0.05). OTUs with significantly greater abundance in disturbance treatments were then categorized by their distribution patterns (shown in pie charts): OTUs that were unique to that treatment, OTUs that were also abundant in nearby ground soil (Ground OTUs), and OTUs that were abundant in all samples except undisturbed canopy soil (Canopy Inhibited).

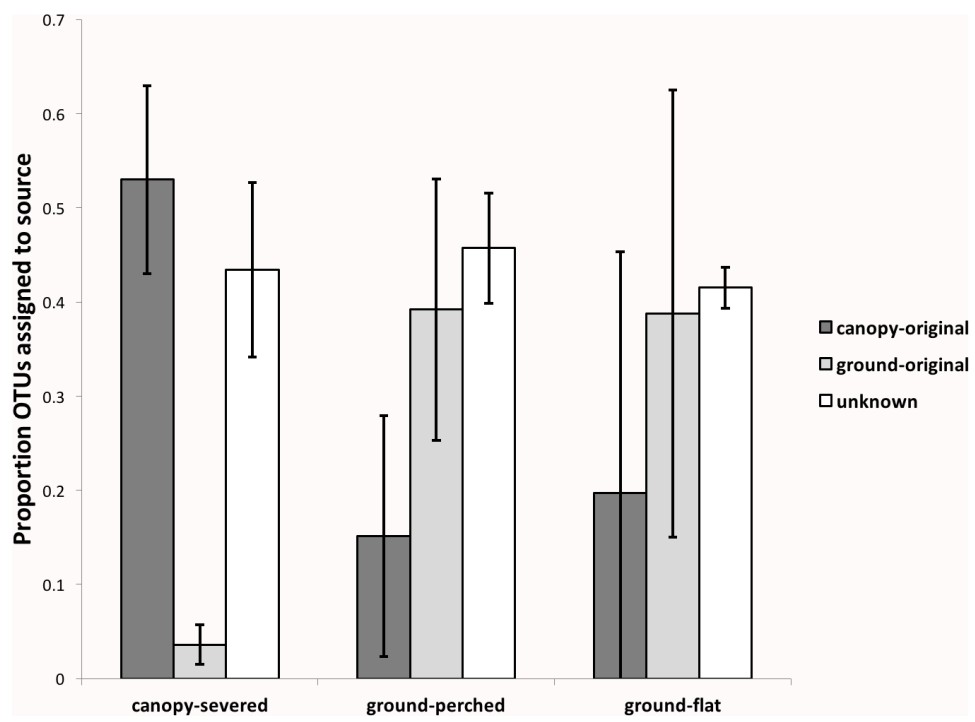

**Figure 6 Proportion of OTUs in each experimental treatment (canopy-severed, ground-perched, and ground-flat) that could be assigned to a canopy (dark gray bars) or ground (light gray bars) source by SourceTracker.** Results reflect the mean among all samples within an experimental treatment, and error bars represent the standard deviation from the mean.

## SourceTracker results

To further investigate how the bacterial communities in the experimental treatments were assembled, we categorized bacterial OTUs according to their likely sources with SourceTracker2. For this analysis, the canopy-original and ground-original samples were considered potential sources, and the experimental treatments were sinks. SourceTracker2 randomly assigns OTUs to the specified sources and calculates the probability that each OTU is derived from its assigned source. These probabilities are used to re-assign each OTU to its most likely source or to an ''unknown'' source, and this process is iterated and repeated until convergence of calculated probabilities is achieved (*Pittl et al., 2010*). Approximately half of the OTUs in canopy-severed treatments could be confidently assigned to a canopy source, while very few OTUs were assigned to ground soil (Fig. 6). In contrast, the ground-perched and ground-flat treatments included many more OTUs assigned to ground-original. Among all treatments, approximately 40% of the OTUs could not be assigned with confidence to either a canopy or a ground source and therefore were assigned to an ''unknown'' source.

## DISCUSSION

### The unique bacterial communities of canopy soil

Canopy soils are presumed to be a harsh environment for most microorganisms. For example, canopy soils are more acidic [canopy pH = 4.6 (*Haristoy, Zabowski & Nadkarni, 2014*); terrestrial pH = 5.4 (*Chandler & Schmidt, 2008*)], and have a higher carbon:nitrogen ratio (*Haristoy, Zabowski & Nadkarni, 2014*). In general, canopy soil temperatures are similar to those on the forest floor throughout the year, but canopy soils can experience short, distinct intervals of "dry-downs" during the dry season, which do not occur on the forest floor (*Aubrey, Nadkarni & Broderick, 2013*). Therefore, the community composition of canopy soils is expected to be distinct from that of ground soil. Canopy and bromeliad leaf-tank soils were shown to be distinct from ground soils according to a fingerprinting analysis (*Pittl et al., 2010*), but no previous studies have investigated temperate rainforests with high-depth environmental sequencing technology. Our results demonstrate that the bacterial communities of canopy soils are distinct from those of ground soils and become more like ground soil when transplanted to the forest floor. However, the difference in community structure is not reflected in a dramatically different bacterial taxonomic composition. Rather, the bacterial taxonomic profile of canopy soils is recognizably similar to that of ground soils. All of the major taxonomic groups of Bacteria found in the soil of the forest floor were also identified in canopy soils, so the distinction between canopy and ground communities must be at a species-like or similarly low taxonomic level.

Given this general taxonomic similarity, the focus of this study was to identify those bacterial "species" (i.e., OTUs = operational taxonomic units) that are unique to the canopy soil and those that responded to experimental disturbances of EM. The most abundant 'missing microbes' of the canopy (i.e., those OTUs that may explain the lower diversity of canopy soil by their absence therein) were identified as a set of "Canopy Inhibited" OTUs that were prevalent in all experimental treatments but not in the original, undisturbed canopy soil. The taxonomic classifications of the "Canopy Inhibited" OTUs are not clearly distinct from the general population. For example, some of the most abundant OTUs belong to the Actinobacteria and Bradyrhizobiaceae, which are also represented in the canopy, but by different and many fewer OTUs. These results are consistent with the "Canopy Inhibited" taxa representing widespread soil bacteria that are unable to thrive in the harsh conditions of the canopy.

### Ground soil bacteria dominate canopy material transplanted to the forest floor

The deposition of canopy EM onto the forest floor appears to trigger a shift in microbial community composition, which could occur via colonization of the EM by nearby ground soil organisms, or by stimulation of organisms that are already present in the canopy EM, or both. Although disentangling cause and effect is not possible with the available data, our results yield insights into the dynamics of bacterial populations in response to disturbances of the canopy EM. First, degradation of canopy EM on the forest floor is accompanied by a replacement of canopy bacteria with typical ground soil bacteria such that the community composition is highly similar to nearby ground soil within two years (Fig. 3). Second, this
transition to a typical ground soil community appears to be unaffected by whether the canopy material is retained on or removed from the branch. Third, our results provide very little evidence that movement of EM to the forest floor stimulated the growth of bacteria that were native to the canopy. Such organisms would have been detected as OTUs with greater abundance in the transplanted material compared to the original canopy soil and also compared to the ground soil. Very few such OTUs were identified (labeled "Unique to Treatment" in Fig. 5). In contrast, the vast majority of OTUs in the transplanted EM could be traced to nearby ground soil by interpretation of the differential abundance results ("Ground OTUs" in Fig. 5).

These results suggest that the accelerated degradation of canopy soils when placed on the forest floor is caused primarily by colonization of the canopy material by nearby ground soil bacteria. However, stimulation of resident canopy bacteria could also play a role, considering that the transplanted materials included OTUs that could not be traced to ground soil, suggesting that the legacy of the canopy is still evident in these samples. Additional work is needed to test whether this is a consistent signal or simply due to incomplete sampling of the environment.

### Severing the connection to the living tree causes distinct shifts in the bacterial community

Canopy soils on branches that were severed from the host tree and suspended in the canopy for two years contained bacterial communities that were distinct from the original canopy community and also from ground soil. These distinctive bacterial communities could have arisen due to dispersal of bacteria from ground soil or from another source not captured by the experimental design. Our results, however, indicate that very few OTUs from the severed canopy EM were derived from ground soil (Figs. 5 and 6), and many of the "Unique to Severed" OTUs were not found anywhere else (Fig. 5). Together, these results point to multiple sources, including those not sampled during this experiment, of organisms that were assembled into the low-diversity and unique community of the severed canopy EM.

Canopy material in the severed branch did not experience accelerated degradation, unlike the material transplanted to the forest floor. However, during visits to the canopy during the study period, EM on severed branches appeared to be drier than EM on intact branches, perhaps because the severed branches could not receive stemflow. These observations, together with the bacterial diversity results, suggest that the severed branches are harsher environments than intact branches of the canopy and that their community composition is the result of the persistence of a subset of the original canopy species plus the colonization of a few opportunistic taxa from elsewhere in the forest.

## CONCLUSIONS

Epiphytic material and associated soils in the canopy constitute large pools of nutrients, water, and carbon in temperate rainforests (*Haristoy, Zabowski & Nadkarni, 2014*; *Matson, Corre & Veldkamp, 2014*). Therefore, the origin and fate of canopy epiphytic material is of central importance to understanding the microbial ecology of temperate rainforests. Our results provide the first in-depth survey of bacterial communities in canopy soils

and reveal them to be taxonomically similar to underlying ground soil but much lower in diversity. The comparatively few bacterial taxa that are highly abundant in canopy soil are distinct members of the same taxonomic groups found in ground soil. Our field experiment demonstrated that the soil created by EM decomposing on the forest floor for two years is nearly, but not completely, indistinguishable from ground soil. However, epiphytic material in the canopy that has been severed from the host tree fosters unique and low-diversity bacterial communities. The bacterial taxa stimulated in the severed branch are derived from multiple sources including the canopy and forest floor, suggesting that they might be exploiting an opportunity to colonize a habitat that has just experienced a massive disturbance. These results highlight the unique nature of canopy-dwelling bacterial communities as well as the importance of the connection to a living tree as an essential component of their canopy ecology.

## ACKNOWLEDGEMENTS

We thank Alex Hyer, Christopher Thornton, Emily Dart, and August Longino for technical assistance with laboratory work and data analyses. Erika Longino, Dennis Aubrey, Autumn Amici, Camila Tejo, Jordan Herman, and Johanna Castillo provided help in the field. Renae Curtz and Doug Cornwall assisted with the illustration in Fig. 1. Jerry Freilich provided research administrative support in the Olympic National Park.

### Funding

The research was funded by grants from the National Science Foundation (DRL 15-14494 and DEB 11-41833) and the University of Utah. There was no additional external funding received for this study. The funders had no role in study design, data collection and analysis, decision to publish, or preparation of the manuscript.

### Grant Disclosures

The following grant information was disclosed by the authors:
National Science Foundation: DRL 15-14494, DEB 11-41833.
University of Utah.

### Competing Interests

The authors declare there are no competing interests.

### Author Contributions

- Cody R. Dangerfield performed the experiments, analyzed the data, wrote the paper, prepared figures and/or tables, reviewed drafts of the paper.
- Nalini M. Nadkarni conceived and designed the experiments, performed the experiments, contributed reagents/materials/analysis tools, wrote the paper, reviewed drafts of the paper.

- William J. Brazelton conceived and designed the experiments, analyzed the data, contributed reagents/materials/analysis tools, wrote the paper, prepared figures and/or tables, reviewed drafts of the paper.

### Field Study Permissions

The following information was supplied relating to field study approvals (i.e., approving body and any reference numbers):

Research permits were issued from the Research Office at the Olympic National Park (OLYM-000234).

### DNA Deposition

The following information was supplied regarding the deposition of DNA sequences:

SRA BioProject PRJNA357844.

### Data Availability

Zenodo: http://doi.org/10.5281/zenodo.597545

GitHub: https://github.com/Brazelton-Lab.

### Supplemental Information

Supplemental information for this article can be found online at http://dx.doi.org/10.7717/peerj.3773#supplemental-information.

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
