# Peer review of "Canopy soil bacterial communities altered by severing host tree limbs"

_PeerJ, doi:10.7717/peerj.3773_

## Round 0.1 · original submission · Major Revisions

The overall manuscript, main questions, and overall experimental design were well received by all three reviewers. However, all three also raised important concerns regarding data analysis and statistical assumptions of the tests employed. I agree with the reviewers that this is one of those cases that falls somewhere between Minor/Major revisions. The experiments were quite sound, interesting, and important, but I would like to see the reviewers' concerns about data analyses clearly addressed upon resubmission.

·

Basic reporting

The manuscript is well written and generally well structured. However, I have some concerns about its current format:

- the current title is unclear. what is the experimental disturbance and what is the effect? perhaps try to capture your main result in you title
-in places more information is needed for the reader to be able to accurately interpret your figures and tables.
The discussion is a little weak and speculative at the present time. Try not to repeat your results to such as extent (ie.. there should not be any need to cite all of your figures in your discussion). Your discussion should, ideally, be reference rich, allowing the reader to understand where your work sites among the broader field of knowledge related to your research question.

Experimental design

This is a fascinating research question and I think that the experimental collection of the data has been well performed. In this regard I don't really have any issue with the experimental design. This research identifies and aims to fill a significant knowledge gap (related to the microbial diversity and distribution of canopy soils and their impact on the wider soil ecosystem).
It would appear that appropriate collection permits were obtained for this study.

Validity of the findings

The main problem I have with the manuscript is that I don't feel the data analysis has been well performed. In my opinion more effort needs to be made to show not only that there are differences in your communities among samples but that these differences are significant (or not). It would appear that you have all of the data that should allow you to conduct more rigorous analyses including pairwise tests of significance for your sample data. Incorporation of some more rigorous statistical tests should allow you to focus your discussion and make it more informative and less speculative.

Additional comments

General comments and suggestions are attached to a PDF of the authors original submission

Reviewer 2 ·

Basic reporting

Dangerfield et al. present a field experiment coupled with 16S amplicon sequencing in order to test the effect of disturbance on canopy soil bacterial communities. They provide useful insights into bacterial dispersal and the drivers of bacterial community composition in a temperate rainforest ecosystem. The manuscript is well written throughout and the authors have made all data, sample information and most of the code available. Overall, the manuscript is mostly sound and I have outlined my concerns below. I found the results lacking in statistical support however and the authors should clarify which methods were used.

Minor point:
Lines 59-66: The motivation for this paragraph was slightly lost on me. Are the authors suggesting that epiphytes die when they fall to the forest floor due to altered environmental factors, one of which is the microbial community? Please clarify.

Figures: Figure 1 is useful to guide the reader regarding experimental design and figure 2 is also useful in showing how the samples cluster together (although the red arrows add little). However, figures 3 and 4 aren’t useful in their current form. Whilst I like the idea of these volcano plots, some explanation is required for the reader to fully interpret their adaptation from methods designed for gene expression studies. What is logCPM? And is it appropriate to still refer to the y-axis as log fold-change?

Experimental design

All seem appropriate. Good to see experimentation in a natural system combined with deep sequencing.

Validity of the findings

Lines 150-158: The authors have made a point of not rarefying their data and refer to the McMurdie and Holmes paper (2014) but then discard samples with less than 20,000 sequences. To me, it seems odd to use diversity measures that account for differences in sequencing depth (Morisita Horn & edgeR) but then throw samples out anyway. My guess is that rarefied samples are required for sourcetracker2 however I felt this analysis added less to the manuscript than the other analyses presented.

I’d be interested to see the impact of this choice not to rarefy the data- perhaps presenting and contrasting an analysis that does rarefy. That may be beyond the scope of the paper but, given the call for such an approach (McMurdie and Holmes, 2014) and relatively few studies that take it, could add some value to the paper.

Lines 158-161: The choice of leaving chloroplast and mitochondrial sequences in the dataset seems to go against the common methods in this field (Mothur MiSeq SOP, 2017). Whilst the authors state their inclusion can “aid ecological interpretation”, the only result presented states that most chloroplast sequences could not be taxonomically classified. The krona plots in the supplementary materials show these sequences are low in relative abundance but I see little value in leaving them in. I’m concerned that their inclusion could skew some of the diversity estimates.


Line 213: “In addition to having lower richness and evenness, canopy soils had significantly different OTU…” but I couldn’t find a description of the statistical methods used to compare beta-diversity. Was some sort of permutational anova used?

Lines 186-197: Similar to the above point, I couldn’t find any tests of significance to support these descriptions. Why not run some GLMs with different diversity metrics?

Lines 205-211: I was interested that similar abundances at the order level but not OTU level were observed- could the authors discuss this further. A link to some ecological theory here could be useful i.e. competition may be strongest among closely related species (see Burns and Strauss, 2011 PNAS for example).

Additional comments

Overall, I think this is a diligent study that is well written. The bioinformatics pipeline used seems sound but the statistical analyses need some clarification in my view. Aside from that all my comments above are suggestions rather than objections to publication.

·

Basic reporting

The article is well written. The diagram in figure 1 helps greatly to understand the design of the experiment and the terminology. A few comments:


line 1 “Trees of temperate rainforests host a large biomass of epiphytes, which are living plants
associated with soils formed in the forest canopy”
>> Increasingly, people use “epiphytes” to mean any organisms living on plants, which could be other plants or could be microorganisms. For me, use of the word epiphytes for microbes is a fair evolution of the word, and I would suggest for clarity in the first sentence: “Trees of temperate rainforests host a large biomass of epiphytic plants, which are associated with soils formed in the forest canopy”. Future use of “epiphytes” is then clear.

>> A real picture of one of the experimental trees would be great, and a picture of the canopy soil. Canopy soil of 5cm.. I’m trying to picture in my head a bunch of potting soil or something sitting on branches to a depth of 5cm… I’m having trouble imagining such a depth of soil on living trees. I understand I can probably google the references but a quick photo would improve the manuscript experience.

line 253 “Chloroplasts and mitochondria (both of which are detected by sequencing of bacterial 16S rRNA genes) were among the most common sources of differentially abundant OTUs between canopy original and ground-original (Figure 3).”
>> Why are the more chloroplast sequences in the canopy? This isn’t necessarily expected, so it needs some explanation.

>> Figure 3 is pretty clear but should have an X axis label. Figure 5 needs a Y axis label

Experimental design

line 125 “The samples were homogenized for DNA extraction by flash-freezing the sample with liquid nitrogen followed by grinding the sample into a fine powder.
>> Please specify how the samples were ground? I guess mortar & pestle?

line 313 “SourceTracker Results”
>> Please explaining briefly how SourceTracker works (in a sentence)

>> It is not clear what the height of the severed branches were. Please add this detail to the methods. Also, did the canopy soil on the ground perched samples come into contact with the forest floor in any way? Was it low enough for rain to splash up from the forest floor? These details are interesting because they provide clues as to how floor bacteria may invade.

You mention “EM on severed branches appeared to be drier than EM on intact branches”.
>> Can you please comment on EM survival across all branches in your study? The introduction explains that survival is often affected by moving to the shade…

Validity of the findings

line 148 “We chose not to cluster sequences any more broadly because clustering inevitably results in a loss of biological information and because no arbitrary sequence similarity threshold can be demonstrated to consistently correspond to species-like units.”
>> 97% similarity is how everybody does it because of 1) sequencing and polymerase error and 2) within a single bacterium multiple rRNA copies can differ by a percent or more. 97% may not be necessary here, but because it’s the norm, please do this in addition to your analysis and report that it makes no substantial difference in the results. Or if it does make a difference, then you can perhaps make an argument that you have real biological information in your samples that is lost by such a standard clustering step, which would be an interesting methodological comment given how standard 97% clustering is.

line 165 “the Morisita-Horn index from a table of OTU abundances across all samples. This index was chosen because it reflects differences in the abundances of shared OTUs without being skewed by unequal numbers of sequences among samples.”
>> does this mean you did not correct for sequencing depth prior to applying the Morisita-Horn index? I have not used this index before but I read that it is highly sensitive to abundant taxa, and therefore is relatively insensitive to undersampling. But insensitivity to undersampling is different than being unaffected by undersampling, no? If this has been done without correcting for sequencing depth, it seems you must reproduce the conclusions using either the rarefied table or a linear model that includes sequencing depth as a variable.

Additional comments

>> Would it be possible to get the soil from the branches and forest floor analyzed for nutrients, such as from an agricultural extension site? Is the nutrient composition of ground perched samples changing to be more like that of forest original, and less like that of canopy original, for example? Seems a potentially important part of the mystery of how the soils become more ground-like.

---

## Round 0.2 · Minor Revisions

Two reviewers have reassessed the newest version of this article and are generally pleased with the changes that have been made. There are still a couple of minor changes that I would like to see made (see in particular Reviewer 1's comments about Table 1 and Figure 6).

For Table 1, I would suggest including a small figure legend underneath that describes what the values and measurements in the table represent. Doesn't have to be elaborate, just something so that the reader can easily parse what they're reading in the table without having to search through the paper.

For Figure 6, it would be good to include a set of basic statistical analyses (ANOVA/Tukey's HSD) in the figure itself. Even if NS.

·

Basic reporting

I feel the manuscript is generally well written and presented

Experimental design

Good - please make sure to include a few more details about your design in the introduction and abstract (i.e. where was you study based and on what type of tree)

Validity of the findings

Good. I note that efforts have been made to improve statistical interpretation of these data. However, data in table 1 still need some additional tests performed to confirm if differences are significant (Turkey test?) - same can be said for the data in figure 6.

Additional comments

Thanks for the opportunity to again review this manuscript. There are still some changes that I feel need to be made. I feel these are quite minor in nature.

Abstract.
Overall this reads well and the objectives and aims are clear. Please add some details on the location of your study (i.e., country/region). It would be good if you could also identify that your study focuses on the canopy soil of Acer macrophyllum as at the moment the scope of your study is not clear from the abstract.

Line 11. I agree that this appears to be among the first studies to look at 16S rRNA from canopy soil. There are studies that have used fingerprinting techniques. You don’t need to add these, but I just raise Plant and Soil 329(1):65-74 for your interest.

Line 15. Can you get rid of the term ‘species level’ and just refer to ‘different taxa’ I don’t think we know that the level you defined your analyses really relates to species.


Introduction
Generally clear and succinct.
Please join paragraphs on lines 33-36 and 38-47. The latter paragraph is not referring to a new subject and does not require a new paragraph
Line 58. Change ‘experimental study’ to experiment.
Again in line 59 you mention you will conduct work in a rainforest but done say where. Maybe you could be more specific, if only to mention ‘in the U.S.A.’

Methods.

FigS1 + Fig 2: Great photos!
Line 131 : change 8 to eight

Line 195. In reference to table 1 it is not appropriate just to say that values were indistinguishable. You need to do the stats on this (Tukey test results added to table 1?). this is important. The need for better stats was raised in your last review


Table 1: what are the +/- values? Is this St Dev or St Err? Please include detail in figure legend.
Results
Line 221-228. I understand that you have done the AMOVA and MDS. You should not treat the MDS and related stats in isololation of eachother. Sure, describe interesting results from the MDS but wherever you see differences in the data and describe them you need to also do stats on these data to confirm the significance of the visual result.

Line 229. You don’t really know if these differences are large. Even small differences can cause clustering on MDS plots if they are consistent among groups. Avoid the word ‘large’.

Figure 6. Remove the box around this graph and the horizontal gridlines.
Discussion. The discussion is appropriate and the conclusions appear well balanced for the study.

Reviewer 2 ·

Basic reporting

Having read the revised manuscript and associated rebuttals I am satisfied with the changes. The introduction and experimental aims are now clearer, as is the explanation of the rationale behind the statistical tests.

Experimental design

This aspect of the study was fine in the first revision. No need for changes.

Validity of the findings

The authors have done a decent job of improving the way the statistical tests were reported and included a few additional analyses. Together these make the overall results more interpretable. There is a vast number of ways of analysing 16S data and the authors have justified their methodology. Both data and code are available.

Additional comments

The manuscript is much improved regarding the clarity of the text, particularly the explanation of statistical tests, and the additional analyses help guide the reader to overall differences among treatments before getting in to the OTU tracking. Clearly no singular method of analyzing 16S data is better than others at present, so the explanations are useful.

---

## Round 0.3 · accepted · Accept

Thank you for submitting the revision, and I'm happy to accept the article.

Apologies again for the mix up with the first annotated manuscript, that's on me and just know that I learned an important editing lesson.